# Qualitative Identification of the Static Pull-In and Fundamental Frequency of One-Electrode MEMS Resonators

**DOI:** 10.3390/mi9120614

**Published:** 2018-11-22

**Authors:** Jianxin Han, Lei Li, Gang Jin, Wenkui Ma, Jingjing Feng, Haili Jia, Dongmei Chang

**Affiliations:** 1Tianjin Key Laboratory of High Speed Cutting and Precision Machining, School of Mechanical Engineering, Tianjin University of Technology and Education, Tianjin 300222, China; jgang1982@tute.edu.cn (G.J.); hljia@tute.edu.cn (H.J.); 16145@tute.edu.cn (D.C.); 2School of Transportation and Vehicle Engineering, Shandong University of Technology, Zibo 255049, China; lleisnowflake@sdut.edu.cn; 3Department of Mechanical Engineering, Henan Mechanical and Electrical Vocational College, Henan 451191, China; mawenkui@tju.edu.cn; 4Tianjin Key Laboratory for Advanced Mechatronic System Design and Intelligent Control, School of Mechanical Engineering, Tianjin University of Technology, Tianjin 300384, China

**Keywords:** MEMS, resonator, one-electrode, static pull-in, fundamental frequency

## Abstract

This paper attempts to qualitatively identify the static pull-in position, pull-in voltage, and fundamental frequency of one-electrode microresonators from a physical perspective. During theoretical derivation, a generalized one-degree-of-freedom (1-DOF) model in nondimensional form derived using the differential quadrature method (DQM) is first introduced and then transformed for frequency normalization. Based on the deduced formulas, the upper and lower bounds of the static pull-in position and pull-in voltage are both deduced through mathematical proof. To distinguish the monotonic and nonmonotonic behavior of the fundamental frequency versus direct current (DC) voltage, a critical condition decided only by cubic stiffness is then determined. For the first time, two extreme static positions, as well as the corresponding fundamental frequencies and DC voltages to identify different frequency behaviors are derived, and their variations versus cubic stiffness are then discussed and verified. During the simulation process, a high-order DQM and COMSOL 2D model are both applied for numerical analyses. Guided by nondimensional results, typical behaviors with specific physical parameters are examined in detail. Results demonstrate that the curve tendencies between all the qualitative results and quantitative numerical simulations in dimensional form agree well with each other, implying the possibility of using 1-DOF model to qualitatively discuss physical parameters effects on the system statics and dynamics.

## 1. Introduction

Electrically-actuated microbeam-based devices have long been applied to design various sensors and actuators in micro-electro-mechanical systems (MEMS) due to their geometric simplicity and broad applicability [1,2]. To acquire better performance, it is crucially important for MEMS designers to grasp the static and dynamic properties of the core components in these devices, among which the static pull-in position, pull-in voltage, and fundamental frequency are undoubtedly the most important design indicators in the MEMS community due to their decisive effects on system performances [3]. Until now, there are still two sustained concerns around these aspects. One is the variation of the static pull-in position and pull-in voltage, which may show some differences among various works in literature [4,5,6,7,8]. The other is the diagram depicting the fundamental frequency versus direct current (DC) voltage, which can exhibit monotonic or nonmonotonic behavior for different system parameters [9,10,11,12].

Investigations on the static pull-in position, pull-in voltage, and fundamental frequency of MEMS resonators can be mainly separated into two groups. The first group focuses on qualitative identification via low dimensional models such as time-varying model [11,13,14], and aims to give some theoretical explanations to the statics and dynamics of the system. The second group is devoted to quantitative investigations via continuous models governed by partial differential equations [15] or via finite element simulation performed with commercial code such as ANSYS [10], COMSOL Multiphysics [16], and so on. Undoubtedly, experimental explorations on such MEMS resonators are the most convincing analyses [17]. However, from predesign perspective, theoretical or numerical investigations seem to be more suitable than experimental ones. Based on theoretical or numerical results, one can then fabricate experimental samples and carry out verifications or more in-depth investigations [18,19]. No matter what the investigation process is, the final goal is to provide detailed understanding of the system and give some guidance for system design and optimization. In this paper, we mainly focus on one of the most common MEMS devices, i.e., one-electrode MEMS resonators [16,20]. From qualitative consideration, the static pull-in position, pull-in voltage, and fundamental frequency of the system are investigated in depth via a generalized one degree of freedom (1-DOF) model derived using differential quadrature method (DQM). What follows are some discussions about the static pull-in position, pull-in voltage, and fundamental frequency of this type of microbeam-based resonators.

The time-varying capacitor model has been regarded as an irreplaceable model during static and dynamic analyses of MEMS devices [1,2]. Without the consideration of midplane stretching effect of microbeam, nondimensional pull-in position is 1/3 and pull-in voltage is 4/27 [14]. However, microbeam may undergo relatively large deformation during some working conditions below static or dynamic pull-in instability. An additional tension effect of microbeam induced by midplane stretching cannot be neglected [21]. In this case, the static pull-in position is greater than 1/3, and the pull-in voltage is greater than 4/27. From a quantitative analysis perspective, continuous models via Euler-Bernoulli beam theory or others [22,23,24] seem to be more suitable for system analysis than time-varying capacitor models. Based on different modeling considerations, the static pull-in position may exhibit slight differences in different studies, of which the details can be found in Ref. [3]. Moreover, based on continuous models, one can also consider various physical factors that may affect the static pull-in position and pull-in voltage, such as scale effect [8], fringing effect [12], Van der Waals forces [25], and surface effect [26]. Similarly, the fundamental frequency of MEMS resonators can also be investigated via time-varying capacitor model. If midplane stretching of the microbeam is not considered, the fundamental frequency can only exhibit monotonic behavior versus DC voltage [11]. Actually, midplane stretching exists in the system, and sometimes can induce a nonmonotonic behavior of the fundamental frequency. In many studies, this phenomenon has been observed and discussed [5]. It is totally different from traditional thinking that the fundamental frequency decreases with an increase in DC voltage. The mechanical mechanism continues to attract many scholars [10,27,28].

As far as we know, there are still two specific issues to be determined. They are:

(i) Variation of the static pull-in position and pull-in voltage. It seems to be an old question, as so many studies have been done on this aspect. In the literature, it is shown that the pull-in position is affected by the midplane stretching and some other factors [3,5,10]. Normalized maximum static pull-in position of up to 0.39 can be observed [5]. Sometimes, this threshold is 0.68 [29]. Thus, physical parameter identifications on the static pull-in position, as well as the corresponding pull-in voltage, become necessary during design and optimization procedures of MEMS devices. The most effective way to solve these problems is to mathematically prove the variation properties of both the above parameters, and then grasp the influence of dimensional parameters on them. Perhaps it is more feasible to start these investigations via an appropriate low dimensional model.

(ii) Variation of the fundamental frequency. For different structure parameters, the fundamental frequency of the system can exhibit different variation property with an increase in DC voltage. In addition to monotonic behavior versus DC voltage, a nonmonotonic behavior may also be observed in the system [5,10]. However, the critical conditions to identify the existence of this phenomenon are still unclear. Out of curiosity, we want to identify the effect of physical parameters on the switch pattern of these phenomena through analytical investigations via a low dimensional model, and finally, draw some qualitative conclusions.

In our previous work [30,31,32], a simplified time-varying model is considered whereby the electrostatic force is treated as a concentrated force on microbeam. Equivalent parameters are obtained via traditional Galerkin procedure. Although this model establishes the relationship between nondimensional and dimensional parameters, the concentrated force assumption seems to be not appropriate during the equivalent procedure. At this point, it seems that the 1-DOF model derived using Galerkin decomposition on continuous model can overcome this shortcoming. However, during the integration procedure, electrostatic terms cannot be analytically solved. Approximations must be used which may induce the disappearance of homoclinic or heteroclinic points (unstable position) in the system [7]. Finally, typical dynamic pull-in property under primary resonance conditions may not be found, which is unreasonable in practice [32]. Thus, the 1-DOF model from the Galerkin method may not be appropriate as well, especially for subsequent dynamic pull-in analysis from a qualitative/quantitative perspective. Fortunately, Najar et al. [22,33] derived a generalized 1-DOF model via DQM, and investigated the statics and dynamics of an electrostatic microactuator based on sets of physical parameters. Results show strong qualitative consistencies compared with a high-order DQM discretized model. Their work shed light on the possibility of describing the relationship between theoretical solutions via a nondimensional 1-DOF model and numerical simulations via a high-order DQM model. Actually, DQM is a well-known method to investigate partial differential equations. This method uses a discrete-point approximation of the deflection such as Chebyshev-Gauss-Lobatto grid points rather than mode-shape decomposition to approximate the response of a flexible structure [33]. In addition to quantitative analysis with specific parameters via a high-order DQM model using two or more grid points on microbeam, it can also qualitatively investigate the statics and dynamics (such as primary resonance) of the system via a 1-DOF model using only one grid point on the microbeam. Compared with previous 1-DOF time-varying capacitor models or the 1-DOF Galerkin model, the advantage of this 1-DOF DQM model is that no hypothesis or approximation on electrostatic terms is introduced. It is just an original, simplified model through spatial discretization of a continuous model, and can qualitatively hold the static and primary resonance properties of the system. Thus, we plan to carry out our investigations based on this generalized 1-DOF model. For frequency normalization purposes, a time transformation is first reintroduced. A new form of dynamic equation is obtained and then applied in all our theoretical investigations.

The structure of this paper is as follows. In Section 2, a continuous model to describe one-electrode microresonators is introduced, and then a generalized 1-DOF model discretized with one grid point is obtained via DQM. Based on the model by Najar et al. [22,33], a dynamic equation of motion with frequency normalization is deduced by reusing time scale transformation. In Section 3, the formulas to determine the static pull-in position, pull-in voltage, and fundamental frequency are derived. Their evolution properties are then proved and, finally, verified through numerical simulations. The fundamental frequency is simulated first, and then the threshold to identify the monotonic and nonmonotonic behaviors is theoretically found for the first time. In Section 4, guided by nondimensional results, high-order DQM solutions (using 11 grid points) and COMSOL 2D simulations are applied to discuss the static pull-in property and fundamental frequency in the physical parameter region. Finally, a discussion and conclusions are presented in Section 5.

## 2. System Description

A microbeam-based resonator in Figure 1 is actuated by an electric load $V_{D}\;+\;V_{A}\sin(\Omega \cdot \hat{t})$, where $\hat{t}$ is the time, $V_{D}$ is the DC voltage, and $V_{A}$ and Ω are the amplitude and frequency of the AC voltage. l, b, and h are the length, width, and thickness of the microbeam, respectively. $g_{0}$ is the initial gap width between the microbeam and the stationary electrode. This microbeam is doubly clamped which can induce midplane stretching during static deformation or dynamic vibration.

Many studies have been done on this type of microresonators, providing lots of valuable references on the static pull-in position, pull-in voltage, and fundamental frequency. From the application perspective, some studies focus on nondimensional analysis [5], while some focus on dimensional analysis through using detailed physical parameters [34]. Few works focus on comparative analyses between 1-DOF model and the continuous model from a qualitative and physical viewpoint. To this end, we try to grasp the variation properties of the static pull-in position, pull-in voltage, and fundamental frequency through mathematical proof and verification. All we want is to provide some theoretical design ideas for the optimization of real MEMS devices.

### 2.1. Continuous Model

Considering the midplane stretching of microbeam, the equation of motion of the system in Figure 1 can be written as [10,16,20,35]
(1)$$\tilde{E}I\frac{\partial^{4}\hat{w}}{\partial x^{4}}+\hat{c}\frac{\partial \hat{w}}{\partial \hat{t}}+\rho_{s}bh\frac{\partial^{2}\hat{w}}{\partial {\hat{t}}^{2}}=[P_{0}+\frac{\tilde{E}bh}{2l}\int_{0}^{l}{\left(\frac{\partial \hat{w}}{\partial x}\right)}^{2}\mathrm{dx}]\frac{\partial^{2}\hat{w}}{\partial x^{2}}+\frac{\varepsilon_{0}\varepsilon_{r}b{\left[V_{D}+V_{A}\sin(\Omega \hat{t})\right]}^{2}}{2{\left(g_{0}-\hat{w}\right)}^{2}}\;$$
with the following boundary conditions
(2)$$\hat{w}(0,\hat{t})=\hat{w}(l,\hat{t})=0,\;\frac{\partial \hat{w}(0,\hat{t})}{\partial x}=\frac{\partial \hat{w}(l,\hat{t})}{\partial x}=0\;$$
where x is the position along the microbeam, $\hat{w}(x,\hat{t})$ is the transverse displacement of the microbeam. $\tilde{E}$ is the effective Young’s modulus with $\tilde{E}=E/\left(1-\upsilon^{2}\right)$ for a wide microbeam (b≥5h) and $\tilde{E}=E$ for a narrow microbeam (b<5h), in which E is the Young’s modulus and υ is the possion’s ratio. $\hat{c}$ is the damping coefficient per unit length. I is the moment of inertia of the crosssection with $I=bh^{3}/12$. $\rho_{s}$ is the density. $\varepsilon_{0}$ and $\varepsilon_{r}$ are the absolute and relative dielectric constants, respectively. $P_{0}$ is the effective axial tension force induced by residual stress, temperature variation while in use or predesign process [36].

Buckling condition of this doubly clamped microbeam, i.e., Euler equation, can be written as
(3)$$P_{0,c}^{\mathrm{Euler}}=-\frac{4\pi^{2}\tilde{E}I}{l^{2}}\;$$
For convenience, we introduce the following nondimensional variables
(4)$$w=\frac{\hat{w}}{g_{0}},\;\xi =\frac{x}{l},\;t=\frac{\hat{t}}{T_{0}}\;$$
where $T_{0}=\sqrt{\rho_{s}bhl^{4}/\left(\tilde{E}I\right)}$, then the nondimensional dynamic equation of motion can be given by
(5)$$\frac{\partial^{4}w}{\partial \xi^{4}}+c\frac{\partial w}{\partial t}+\frac{\partial^{2}w}{\partial t^{2}}=[P+\alpha_{1}\int_{0}^{1}{\left(\frac{\partial w}{\partial \xi }\right)}^{2}d\xi ]\frac{\partial^{2}w}{\partial \xi^{2}}+\frac{\alpha_{2}{\left[V_{D}+V_{A}\sin(\omega t)\right]}^{2}}{{\left(1-w\right)}^{2}}\;$$
with boundary conditions
(6)$$w(0,t)=w(1,t)=0,\;\frac{\partial w(0,t)}{\partial \xi }=\frac{\partial w(1,t)}{\partial \xi }=0\;$$
where $c=\frac{\hat{c}l^{4}}{\tilde{E}IT_{0}}$, $P=\frac{P_{0}l^{2}}{\tilde{E}I}$, $\alpha_{1}=6{\left(\frac{g_{0}}{h}\right)}^{2}$, $\alpha_{2}=\frac{6\varepsilon_{0}\varepsilon_{r}l^{4}}{\tilde{E}h^{3}g_{0}^{3}}$, $\omega =\Omega T_{0}$.

Then, buckling condition in nondimensional form can be written as
(7)$$P_{c}^{\mathrm{Euler}}=-4\pi^{2}\;$$

### 2.2. A Generalized One Degree of Freedom (1-DOF) Model Via Differential Quadrature Method (DQM)

For the above type of microresonator, a generalized 1-DOF model via DQM by using one grid point (midpoint) on microbeam can be expressed as [22]
(8)$${\ddot{w}}_{3}+c{\dot{w}}_{3}+\omega_{0}^{2}w_{3}+\gamma_{0}w_{3}^{3}=\frac{\alpha_{2}{\left[V_{D}+V_{A}\sin(\omega t)\right]}^{2}}{{\left(1-w_{3}\right)}^{2}}\;$$
where $\omega_{0}=\sqrt{384+16P}$ and $\gamma_{0}=85.33\alpha_{1}$. $w_{3}$ is the displacement of the midpoint of microbeam, $\omega_{0}$ represents the fundamental frequency of the microresonator without DC voltage, which should be greater than zero below static instability. $\gamma_{0}$ represents the cubic stiffness induced by the midplane stretching. Detailed discrete process can be found in [22].

The generalized buckling condition can be estimated by using the static instability condition $\omega_{0}=0$, which can be finally given by
(9)$$P_{c}^{\mathrm{DQM}}=-24\;$$

Najar et al. [22] verified that both the static and dynamic solution curves by solving Equation (8) have similar tendencies to those via a high-order DQM model. Based on this 1-DOF model, they qualitatively investigated the dynamics and global stability of a microbeam-based electrostatic microactuator. Jumping motion and safe basin of attraction were determined and discussed in depth via a detailed set of physical parameters. Their works are excellent and meaningful for parameter design of this MEMS device, which motivates us to qualitatively identify the static pull-in property and fundamental frequency of this system. To realize frequency normalization, here we reintroduce a time scale transformation $\tau =\omega_{0}t$ on Equation (8). Then, a new form of nondimensional equation of motion can be obtained as
(10)$$\frac{d^{2}w}{d\tau^{2}}+\mu \frac{dw}{d\tau }+w+\alpha w^{3}=\frac{\gamma }{{\left(1-w\right)}^{2}}+\frac{2\gamma \rho \sin(\omega_{e}\tau )}{{\left(1-w\right)}^{2}}+\frac{\gamma \rho^{2}\sin^{2}(\omega_{e}\tau )}{{\left(1-w\right)}^{2}}\;$$
where $w=w_{3}$, $\mu =\frac{c}{\omega_{0}}$, $\alpha =\frac{\gamma_{0}}{\omega_{0}^{2}}$, $\gamma =\frac{\alpha_{2}V_{D}^{2}}{\omega_{0}^{2}}$, $\rho =\frac{V_{A}}{V_{D}}$ and $\omega_{e}=\frac{\omega }{\omega_{0}}$. According to Equation (10), one can notice that α is a generalized nondimensional cubic stiffness of the microbeam and γ is a generalized nondimensional DC voltage. The following analysis will focus on these two nondimensional parameters.

Without consideration of midplane stretching (α=0), we can easily derive from Equation (10) that the static pull-in position $w_{p}=1/3$ and the pull-in voltage $\gamma_{p}=4/27$. If midplane stretching is considered, then $w_{p}>1/3$ and $\gamma_{p}>4/27$. Najar et al. [33] discussed this difference through some analytical analyses. However, variation properties of the static pull-in as well as the fundamental frequency were not discussed in detail. Here, we attempt to figure out this problem.

## 3. Qualitative Identification

In this section, theoretical investigations on the static pull-in position, pull-in voltage, and fundamental frequency of microresonator will be carried out in depth based on 1-DOF model Equation (10).

### 3.1. Static Pull-In Position and Pull-In Voltage

From Equation (10), one can obtain the static position equation as follows
(11)$$\Phi (\alpha ,\gamma ,w_{e})=w_{e}+\alpha w_{e}^{3}-\frac{\gamma }{{\left(1-w_{e}\right)}^{2}}=0\;$$
where $w_{e}$ represents the static position, a function of cubic stiffness α and DC voltage γ.

Equation (11) is not a traditional equation and the static position cannot be determined analytically. However, we can analytically discuss the pull-in property with it. The static pull-in voltage can be determined through the following implicit differentiation equation
(12)$$\frac{\partial \gamma }{\partial w_{e}}=-\frac{\partial \Phi /\partial w_{e}}{\partial \Phi /\partial \gamma }=0\;$$
Finally, the nondimensional static pull-in voltage $\gamma_{p}$ can be derived as
(13)$$\gamma_{p}=\frac{1}{2}{\left(1-w_{p}\right)}^{3}(1+3\alpha w_{p}^{2})\;$$
where $w_{p}$ is the nondimensional static pull-in position, a special static position $w_{e}$.

Substituting Equation (13) into Equation (11), one can derive that the static pull-in position $w_{p}$ can be decided by the following equation
(14)$$\Theta (\alpha ,w_{p})=5\alpha w_{p}^{3}-3\alpha w_{p}^{2}+3w_{p}-1=0\;$$

The above equation is a cubic algebra equation about $w_{p}$ and has theoretical solutions. As different cubic stiffness α may yield different real roots of Equation (14), the first step is to discuss the discriminant of Equation (14), which can be given by
(15)$$\Delta_{1}=\frac{4+{\left(\alpha -1\right)}^{2}}{625\alpha^{3}}\;$$

As cubic stiffness α is positive in practice, discriminant $\Delta_{1}$ is always more than zero. Then, Equation (14) has only one real root and two conjugate complex roots. According to the relationship between roots and parameters, one can easily conclude that the real root is positive. Thus, the static pull-in position $w_{p}$ can be given by
(16)$$w_{p}=\frac{1}{5}\left[1+\sqrt[{3}]{1+\frac{5}{\alpha }-5\sqrt{\frac{5\;+\;(\alpha \;-\;2)\alpha }{\alpha^{3}}}}+\sqrt[{3}]{1+\frac{5}{\alpha }+5\sqrt{\frac{5\;+\;(\alpha -2)\alpha }{\alpha^{3}}}}\right]\;$$

It is obvious from Equations (13) and (16) that the static pull-in position $w_{p}$ and pull-in voltage $\gamma_{p}$ are determined only by cubic stiffness α. Figure 2 depicts a general understanding of the evolution of the static position $w_{e}$ versus DC voltage γ for different cubic stiffness α. With the increase of α, the static pull-in position $w_{p}$ corresponding to the static pull-in point P $\left(\gamma_{p},w_{p}\right)$ increases rapidly at first and then closes to a upper limitation, while the static pull-in voltage $\gamma_{p}$ keeps growing and seems to have no upper limitation. To explore the phenomena in detail, derivation and proof are done in the following study.

Static pull-in position $w_{p}$ and pull-in voltage $\gamma_{p}$ are determined only by cubic stiffness α. If one can derive the extreme value of $w_{p}$ and $\gamma_{p}$ to α, the upper and lower bounds of $w_{p}$ and $\gamma_{p}$ can then be grasped. However, $w_{p}$ decided by Equation (16) is complex and difficult to analyze. Fortunately, $w_{p}$ also satisfies Equation (14). Based on the implicit differentiation of Equation (14), one can obtain the derivative of $w_{p}$ to α
(17)$$\frac{dw_{p}}{d\alpha }=-\frac{\partial \Theta /\partial \alpha }{\partial \Theta /\partial w_{p}}=\frac{5{\left(3-5w_{p}\right)}^{2}w_{p}^{3}}{6[{\left(5w_{p}-2\right)}^{2}+1]}\;$$
Based on Equations (13) and (14), one can also derive the derivative of $\gamma_{p}$ to α
(18)$$\frac{d\gamma_{p}}{d\alpha }=\frac{\partial \gamma_{p}}{\partial w_{p}}\frac{\partial w_{p}}{\partial \alpha }+\frac{\partial \gamma_{p}}{\partial \alpha }=w_{p}^{3}{\left(1-w_{p}\right)}^{2}\;$$

Equations (17) and (18) are always no less than zero, which indicates that $w_{p}$ and $\gamma_{p}$ monotonically increase with the increase of α. When α theoretically approaches to positive infinity, an upper bound of $w_{p}$ can be deduced based on Equation (16). Then, a corresponding upper bound of $\gamma_{p}$ can also be derived via Equation (13). Similarly, if we do not consider the midplane stretching effect (α=0), a lower bound of $w_{p}$ and the corresponding $\gamma_{p}$ can be obtained based on Equations (11) and (13). For simplicity and clarity, the final solutions are listed in Table 1. In Figure 3, we again simulate the static position and voltage in detail. From this figure, one can notice that our theoretical analysis is correct.

The above static pull-in properties can qualitatively offer some design ideas for those quasi-static MEMS devices such as microswitches. For some dynamic MEMS devices such as dynamical switching devices, resonant sensors and filters, the fundamental frequency is also a crucial design parameter, as many MEMS resonance-based devices are designed according to the principle of primary resonance. Thus, it is also important to grasp system frequency properties. What follows are analytical investigations of the fundamental frequency of the microresonator.

### 3.2. Fundamental Frequency

According to Equation (10), the fundamental (angular) frequency $\omega_{n}$ can be given by
(19)$$\omega_{n}=\sqrt{1+3\alpha w_{e}^{2}-\frac{2\gamma }{{\left(1\;-\;w_{e}\right)}^{3}}}\;$$

Evolutions of the fundamental frequency $\omega_{n}$ are depicted in Figure 4. If cubic stiffness α is small enough, fundamental frequency $\omega_{n}$ decreases with the increase of DC voltage γ. When α≥4, frequency curve contains two extreme point C_1_
$\left(\gamma_{1},\omega_{n,1}\right)$ and C_2_
$\left(\gamma_{2},\omega_{n,2}\right)$. When $\gamma <\gamma_{1}$, $\omega_{n}$ decreases with the increase of γ. When $\gamma_{1}<\gamma <\gamma_{2}$, $\omega_{n}$ increases with the increase of γ. When $\gamma >\gamma_{2}$, $\omega_{n}$ decreases dramatically with the increase of γ until $\omega_{n}=0$, then static pull-in instability is triggered. Besides, with the increase of α, point *C*_1_ approaches to (0,1) while $\gamma_{2}$ and $\omega_{n,2}$ corresponding to point *C*_2_ both increases and seems to have no upper bound.

Through qualitative observation and analysis, we can judge that there must be a critical cubic stiffness $\alpha_{c}$ in system. When $\alpha <\alpha_{c}$, the frequency curve contains no extreme point. When $\alpha =\alpha_{c}$, there is only one extreme point. If $\alpha >\alpha_{c}$, there must be two extreme points. If the critical value $\alpha_{c}$ can be derived, then the variations of $\omega_{n}$ can be grasped in depth, which may be useful for MEMS designers.

For a specific microresonator, cubic stiffness α is a specific value. Only DC voltage γ can tune the fundamental frequency $\omega_{n}$. Thus, the derivative of $\omega_{n}$ to γ can be deduced to discuss the variation properties of $\omega_{n}$ versus γ. At extreme point C_1_ and C_2_, based on Equations (11) and (19), the derivative of $\omega_{n}$ to γ satisfies
(20)$$\frac{\partial \omega_{n}}{\partial \gamma }=\frac{\partial \omega_{n}}{\partial w_{e}}\frac{\partial w_{e}}{\partial \gamma }=\frac{5\alpha w_{e}^{3}-9\alpha w_{e}^{2}+3\alpha w_{e}-1}{{\left(1-w_{e}\right)}^{5/2}{\left(1-3w_{e}+3\alpha w_{e}^{2}-5\alpha w_{e}^{3}\right)}^{3/2}}=0\;$$

Based on Equation (14) and the static position property, one can notice that the denominator of Equation (20) is nonzero except the static pull-in position. Before static pull-in instability happens, the numerator of Equation (20) can be used to determine the extreme value of $\omega_{n}$ to γ.

To examine the extreme value of $\omega_{n}$ to γ, we need to introduce a new function, i.e., the numerator of Equation (20), which can be defined by
(21)$$ℜ(\;\alpha ,w_{e}\;)=5\alpha w_{e}^{3}-9\alpha w_{e}^{2}+3\alpha w_{e}-1=0\;$$

The roots of the above equation correspond to the static position on point C_1_ and C_2_. After some derivations, one can obtain the roots of Equation (21) as follows
(22)$$w_{e,1}=\varpi \cdot \sqrt[{3}]{-\frac{q}{2}+\sqrt{{\left(\frac{q}{2}\right)}^{2}+{\left(\frac{p}{3}\right)}^{3}}}+\varpi^{2}\cdot \sqrt[{3}]{-\frac{q}{2}-\sqrt{{\left(\frac{q}{2}\right)}^{2}+{\left(\frac{p}{3}\right)}^{3}}}+\frac{3}{5}\;$$
(23)$$w_{e,2}=\varpi^{2}\cdot \sqrt[{3}]{-\frac{q}{2}+\sqrt{{\left(\frac{q}{2}\right)}^{2}+{\left(\frac{p}{3}\right)}^{3}}}+\varpi \cdot \sqrt[{3}]{-\frac{q}{2}-\sqrt{{\left(\frac{q}{2}\right)}^{2}+{\left(\frac{p}{3}\right)}^{3}}}+\frac{3}{5}\;$$
(24)$$\;w_{e,3}=\sqrt[{3}]{-\frac{q}{2}+\sqrt{{\left(\frac{q}{2}\right)}^{2}+{\left(\frac{p}{3}\right)}^{3}}}+\sqrt[{3}]{-\frac{q}{2}-\sqrt{{\left(\frac{q}{2}\right)}^{2}+{\left(\frac{p}{3}\right)}^{3}}}+\frac{3}{5}\;$$
where $p=-\frac{12}{25}$, $q=-\frac{9}{125}-\frac{1}{5\alpha }$, $\varpi =\frac{-1+\sqrt{3}i}{2}$ and $i=\sqrt{-1}$.

In addition, the discriminant of Equation (21) can be written as
(25)$$\Delta_{2}=\frac{25+18\alpha -7\alpha^{2}}{2500\alpha^{2}}\;$$

According to Equation (25), one can derive a critical cubic stiffness as
(26)$$\alpha_{c}=\frac{25}{7}\;$$

When cubic stiffness $\alpha =\alpha_{c}$, we can derive that $w_{e,1}=w_{e,2}=1/5$ and $w_{e,3}=7/5$. When $\alpha <\alpha_{c}$, only one real root $w_{e,3}$ exists in Equation (21). We can prove that $w_{e,3}$ monotonously decreases versus α. When $\alpha >\alpha_{c}$, both $w_{e,1}$ and $w_{e,3}$ monotonously decrease versus α. In contrast, $w_{e,2}$ monotonously increases versus α. We can prove that the extreme position $w_{e,3}$ is always greater than one, which is physically impossible in system. Only $w_{e,1}$ and $w_{e,2}$ are physically meaningful. Thus, we mainly focus on $w_{e,1}$ and $w_{e,2}$ in the following research.

The DC voltage γ and the fundamental frequency $\omega_{n}$ at extreme point C_1_ and C_2_ are two more key parameters to be determined. When $\alpha =\alpha_{c}$, we can derive based on Equations (11) and (19) that $\gamma_{1}=\gamma_{2}=128/875$ and $\omega_{n,1}=\omega_{n,2}=\sqrt{6/7}$. When $\alpha >\alpha_{c}$, we can prove that $\gamma_{1}$ monotonously decreases versus α, while $\gamma_{2}$ monotonously increases versus α. $\omega_{n,1}$ and $\omega_{n,2}$ both monotonously increase versus α. After some derivations, we can summarize the variation properties of the extreme position, fundamental frequency, and DC voltage in Table 2. Here, note that all the above proofs can be found in Appendix A.

Now, we need to depict some figures to verify our predictions and solutions above. The final results are shown in Figure 5. From this figure, we can observe when cubic stiffness $\alpha <\alpha_{c}$, there is no extreme point on the fundamental frequency curve. When $\alpha =\alpha_{c}$, the extreme point C_1,2_ emerges, i.e., degenerated extreme point of C_1_ and C_2_. When $\alpha >\alpha_{c}$, minimum point C_1_ and maximum point C_2_ can be observed on the frequency curve. It is obvious from Figure 5 and the above analyses that $w_{e,1}$ and $w_{e,2}$ correspond to extreme point C_1_ and C_2_, respectively. This is an important note which will be used in our next investigation.

It can also be observed from Figure 5c that C_1_ and C_2_, combined with start point O and pull-in point P, can separate the static position curve and the fundamental frequency curve into three regions, Region-I, Region-II, and Region-III. In Region-I, with the increase of DC voltage γ, the fundamental frequency $\omega_{n}$ decreases slightly. In Region-II, $\omega_{n}$ monotonously increases with the increase of γ. In Region-III, with the increase of γ, $\omega_{n}$ decreases until static pull-in instability occurs. Different from fundamental frequency $\omega_{n}$, static position $w_{e}$ always increases with the increase of γ.

Evolutions of the static position, fundamental frequency, and DC voltage of point C_1_ and C_2_ versus cubic stiffness can be observed in Figure 6. From this figure, one can notice that our predictions in Table 2 coincide with numerical results, which verifies the correctness of our theoretical analysis.

In this section, we investigate evolution properties of fundamental frequency $\omega_{n}$ versus nondimensional cubic stiffness α. However, it seems that in engineering applications, the effects of physical parameters on the above phenomena are more meaningful and acceptable than nondimensional diagrams. Guided by the above qualitative results, dimensional analyses must urgently be carried out.

## 4. Numerical Simulation

In this section, high-order DQM [22] using certain number of grid points on microbeam is applied to simulate Equation (5) while COMSOL 2D model [16,37] is also used to simulate the static position and the fundamental frequency of a specific MEMS resonator. Nondimensional results obtained via 1-DOF model Equation (10) are applied for theoretical guidance.

Before carrying out numerical analysis, it is necessary to announce that the relationship between nondimensional parameters and dimensional ones. Here, we consider that the axis force is induced only by the residue stress $\sigma_{0}$. Then, the effective axis force $P_{0}$ can be written as [10]
(27)$$P_{0}=(1-\upsilon )\sigma_{0}bh\;$$

From the above theoretical analyses on the static pull-in and fundamental frequency, one can notice that the nondimensional cubic stiffness α and the DC voltage γ are two key design parameters. Combined with nondimensional processes, we can rewrite these nondimensional parameters as follows
(28)$$\alpha =\frac{2.67\tilde{E}g_{0}^{2}}{2\tilde{E}h^{2}+(1-\upsilon )\sigma_{0}l^{2}}\;$$
(29)$$\gamma =\frac{\varepsilon_{0}\varepsilon_{r}l^{4}V_{D}^{2}}{32g_{0}^{3}h[2\tilde{E}h^{2}+(1-\upsilon )\sigma_{0}l^{2}]}\;$$

Obviously, α is affected by $\tilde{E}$, h, l, $g_{0}$, υ and $\sigma_{0}$, while γ is a function of $\varepsilon_{0}$, $\varepsilon_{r}$, $\tilde{E}$, h, l, $g_{0}$, υ, $\sigma_{0}$, and $V_{D}$. Thus, α and γ is not totally independent with each other. Variation of α must induce the variation of γ. Thus, to some extent, the above nondimensional diagrams seem to be only available for general understanding of the statics and dynamics of system. Guided by these qualitative results, we can now investigate the effects of different physical parameters on the static pull-in position, pull-in voltage, and fundamental frequency of the system.

### 4.1. Convergence Analysis

A specific microbeam-based resonator is used during the following numerical simulation [35]. The geometric and material parameters are listed in Table 3. MEMCAD 3D results in [38] is also applied for convergence verification. Table 4 shows the static pull-in voltage via DQM and COMSOL 2D model. Compared with MEMCAD 3D results, one can notice that COMSOL 2D model is available for this MEMS resonator analysis. Besides, when grid points *n* are equal to 11, DQM results are of high accuracy, high convergence stability, and relatively low time consumption. Thus, we select *n* = 11. Detailed observations are shown in Figure 7. DQM results with 11 grid points show excellent agreement with COMSOL 2D model, which satisfies the convergence precision.

To be more convincing, we also compare our DQM calculated results (11 grid points) with calculated and experimental results in [9,39]. Detailed parameters can be found in [39]. From compared results in Table 5 and Figure 8, one can notice that our present DQM calculations with 11 grid points have relatively high prediction accuracy, and this solution procedure can be used to investigate the fundamental frequency and pull-in voltage of the system.

### 4.2. Static Pull-In Analysis

Static pull-in property is discussed first. Beam length *l* is set to be 350 μm. Initial gap width $g_{0}$ is set to be 1 μm and kept constant during analysis. Then, under buckling condition, we can calculate the dimensional residue stress is $\sigma_{0}^{\mathrm{DQM}}=-26.5132\;\mathrm{MPa}$ for the generalized 1-DOF model Equation (10) and $\sigma_{0}^{\mathrm{Euler}}=-43.6124\;\mathrm{MPa}$ for the continuous model Equation (1). Here, we only investigate the residue stress effects on the static pull-in property of the system.

Figure 9 shows the qualitative results depicting the variations of the static position $g_{0}\cdot w_{e}$ versus the residue stress $\sigma_{0}$. Detailed understanding of the dimensional static pull-in position $g_{0}\cdot w_{p}$ and pull-in voltage $V_{D,p}$ can be observed in Figure 10. From this figure, one can notice that $g_{0}\cdot w_{p}$ is close to 3/5 μm when $\sigma_{0}$ approaches to the buckling condition $\sigma_{0}^{\mathrm{DQM}}$, then decreases dramatically, and finally tends to 1/3 μm, while the pull-in voltage $V_{D,p}$ increases from 3.36 V and seems to have no upper bound. This characteristic shows obvious differences compared with nondimensional results in Figure 2 and Figure 3. Actually, the increase of $\sigma_{0}$ can decrease the value of nondimensional cubic stiffness α Equation (28). Meanwhile, it can also influence the nondimensional DC voltage γ Equation (29). $V_{D}$ and $\sigma_{0}$ can both affect the real value of γ. Thus, we observe the above differences. We can also explain the above characteristics in Figure 9 and Figure 10 from a physical perspective. The increase of residue stress $\sigma_{0}$ can induce a reduction of pressure or an increment of tension applied on microbeam. Then, a strengthened effect which can resist deformation of the microbeam becomes more and more obvious. This finally yields much smaller pull-in positions but much larger pull-in voltages.

The above results imply that analytical conclusions summarized only by nondimensional diagrams may have some obvious limitations. Guided by nondimensional analyses, dimensional analyses must also be carried out.

Next, guided by the qualitative results, we use high-order DQM model with 11 grid points to investigate the residue stress effects on the static pull-in position and pull-in voltage. Similarly, we start our numerical simulation with the residue stress $\sigma_{0}\approx \sigma_{0}^{\mathrm{Euler}}$. Figure 11 shows the DQM results depicting the static position versus DC voltage with the increase of residue stress, in which COMSOL 2D results are provided to illustrate the correctness of numerical simulations. Figure 12 exhibits the detailed variations of the static pull-in position and pull-in voltage versus the residue stress. Relative errors calculated according to (DQM results–COMSOL results)/(COMSOL results) × 100% are also shown in Figure 12c. It is obvious from this figure that the DQM results agree well with COMSOL 2D results except some values when $\sigma_{0}$ is close to buckling residue stress. Perhaps the reason is that numerical solutions are more sensitive to DC voltage when buckling instability is about to be triggered. Note that for this microresonator, the maximum static pull-in position is approximately 0.65 μm, and minimum pull-in position is approximately 0.39 μm, while the minimum static pull-in voltage is approximate to 3.37 V, and it seems to have no maximum pull-in voltage. Here, we are more concerned with the curve tendencies between 1-DOF model and the continuous model. Through comparison of Figure 9 and Figure 11, we find that this generalized 1-DOF model can totally reflect the influences of residue stress on the static pull-in properties. Curve tendencies in Figure 10 and Figure 12 also show excellent agreement with each other.

### 4.3. Fundamental Frequency Analysis

To discuss the dimensional fundamental frequency, the beam length *l* is set to be 350 μm and the residue stress $\sigma_{0}=-25\;\mathrm{MPa}$. Only the effect of initial gap width $g_{0}$ is considered. Figure 13 exhibits the qualitative property of the fundamental frequency $\omega_{0}\cdot \omega_{n}/\left(2\pi \cdot T_{0}\right)$ versus DC voltage $V_{D}$ for different initial gap width $g_{0}$. An interesting phenomenon is observed that the extreme point C_1_ and C_2_ cannot hold the same variation properties as the nondimensional diagram in Figure 4. That is because the initial gap width $g_{0}$ can influence both the nondimensional cubic stiffness α and the DC voltage γ. An increase of $g_{0}$ can increases α, while decreases γ, inducing such differences between dimensional diagrams and nondimensional ones. Quantitative results via DQM and COMSOL 2D model can also be observed in Figure 14. Compared with Figure 13 and Figure 14, one can notice that the curves show a consistent trend with each other as well. These demonstrate that dimensional analysis via the 1-DOF model can be used for qualitative identification of the fundamental frequency of the system.

Next, the static positions, fundamental frequencies and DC voltages corresponding to the extreme point C_1_ and C_2_ are discussed in detail. Qualitative results via the 1-DOF model are shown in Figure 15 and Figure 16. We can observe that the static positions in Figure 15a and Figure 16a, the fundamental frequencies in Figure 15b show similar trends compared with Figure 6a,b, respectively. However, the fundamental frequencies in Figure 16b, the DC voltages in Figure 15c and Figure 16c have some differences compared with nondimensional diagrams in Figure 6b,c, respectively. The reason is that the initial gap width $g_{0}$ can affect the cubic stiffness α and the DC voltage γ both. These also imply that nondimensional diagrams may ignore some nondimensional parameter coupling effects on the system statics and dynamics. Figure 17 and Figure 18 show the quantitative properties of the above three physical parameters corresponding to extreme point C_1_ and C_2_. From the results, one can notice that all the variation properties have similar trends as those in Figure 15 and Figure 16. The errors between DQM and COMSOL 2D model seem to be acceptable, which verifies the correctness of our analytical and numerical results.

## 5. Discussion and Conclusions

In the present work, we present some qualitative investigations of one-electrode microresonators via a generalized 1-DOF model derived using DQM (with one grid point). Variations of the static pull-in position, pull-in voltage, fundamental frequency, and in particular, the extreme points on the fundamental frequency curve are deduced and then proved using mathematical methods which can give a global understanding of the statics and dynamics of the system. This led us to undertake some typical numerical simulations guided by nondimensional conclusions via 1-DOF model. Variation properties of some crucial parameters are investigated through comparisons between the qualitative results via 1-DOF model and the quantitative results via high-order DQM (with 11 grid points) and COMSOL 2D model. Results demonstrate that all the dimensional diagrams with the 1-DOF model show similar trends to those of quantitative results via high-order DQM and COMSOL 2D simulations. These indicate the possibility of using the 1-DOF model to qualitatively grasp different physical parameter effects on the system statics and dynamics, but with fewer calculations, less time consumption, and most importantly, high design efficiency. 

During the design and optimization of such MEMS devices, one can borrow ideas from the above analytical results. For instance, according to industry demand, one can design specific pull-in voltage and position which may be applied in microswitch field. One can also realize resonance frequency enhancement by making the ordinate of C_2_ greater than that of O (Figure 5), which may have potential in the resonant sensor field. Moreover, for a specific microbeam-based resonator, proper DC voltage can induce linear-like vibration of system [30]. This can increase the linearity of resonant sensors. Note that nonlinear vibration (hardening or softening vibration) can also be applied in MEMS fields such as wide bandwidth MEMS filters. If one wants to grasp the specific vibration of the above resonator, qualitative dynamic analyses need to be carried out in detail. However, this is beyond the scope of our present research.

The above analyses only discuss some typical physical parameters on system properties. The detailed relationships between qualitative and quantitative results are still unclear to some extent. In a word, we can still not grasp the exact “connection bridge” between two types of results. To find their relationships, large amounts of simulation data are needed to find the key connection between 1-DOF results and DQM simulations. If the relationships are grasped, then we can carry out some design analyses on critical physical parameters in depth, and then give verification via experimental results. For instance, we can discuss the critical cubic stiffness $\alpha_{c}$ on the fundamental frequency variation properties through dimensional analysis. Perhaps some valuable conclusions can be found, which is useful for static and dynamic design of MEMS devices. 

Note that all the above qualitatively analyses only discuss the fundamental resonance of the system, as just one grid point (midpoint of microbeam) is considered when using DQM discretization. Actually, this microresonator has many mode shapes such as symmetric and antisymmetric modes, corresponding to different resonance frequencies. For a specific *i*-order modal of the resonator, DC voltage may have a slight effect on mode shape, but no effect on node number and position [28]. It can also induce frequency shift phenomenon which has been reported in [5,27]. Following the present analysis procedure, one can use two or more grid points to discretize continuous model Equation (5), derive two or more degrees of freedom dynamic equation of motion, and carry out study on it. Then, more theoretical results can be obtained which may have positive effects on the design and optimization of some MEMS devices.

## Figures and Tables

**Figure 1 micromachines-09-00614-f001:**
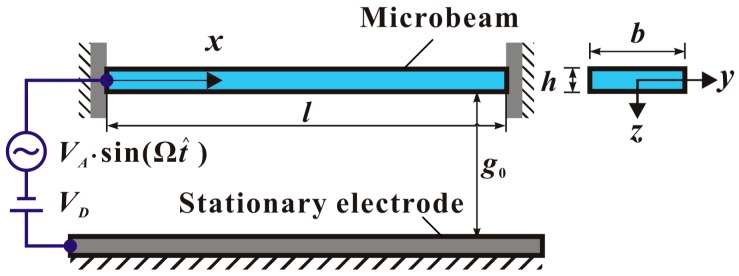
Schematic diagram of a microresonator actuated by one electrode.

**Figure 2 micromachines-09-00614-f002:**
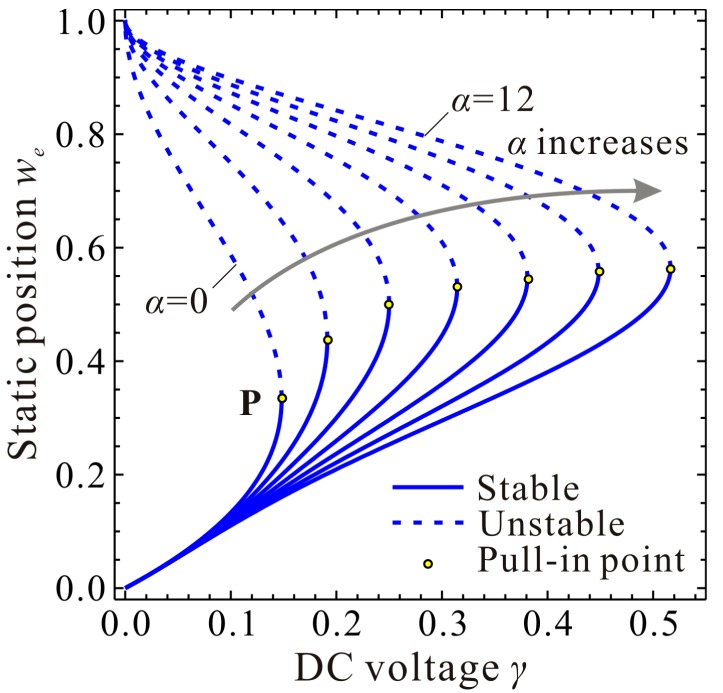
Evolution of the static position $w_{e}$ versus direct current (DC) voltage γ for different cubic stiffness α.

**Figure 3 micromachines-09-00614-f003:**
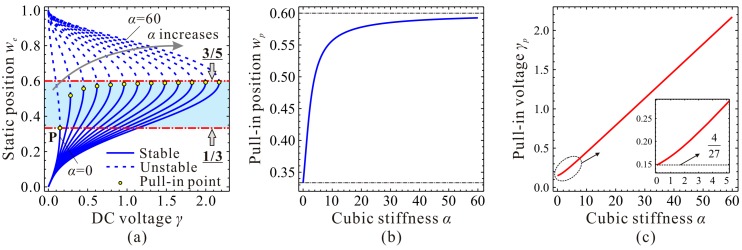
A detailed observation of the static position $w_{e}$ for different cubic stiffness α. global view (**a**); pull-in position $w_{p}$ (**b**); pull-in voltage $\gamma_{p}$ (**c**).

**Figure 4 micromachines-09-00614-f004:**
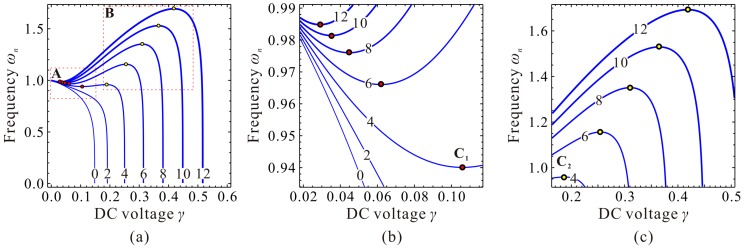
Evolutions of the fundamental frequency $\omega_{n}$ versus DC voltage γ for different cubic stiffness α. global view (**a**); local view in region **A** (**b**); local view in region **B** (**c**).

**Figure 5 micromachines-09-00614-f005:**
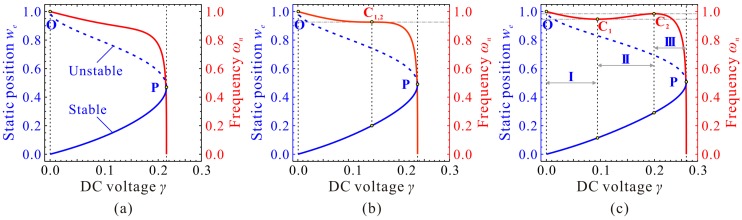
Static position $w_{e}$ and fundamental frequency $\omega_{n}$ for different cubic stiffness α. $\alpha =20/7\;<\;\alpha_{c}$ (**a**); $\alpha =25/7=\alpha_{c}$ (**b**); $\alpha =30/7>\alpha_{c}$ (**c**).

**Figure 6 micromachines-09-00614-f006:**
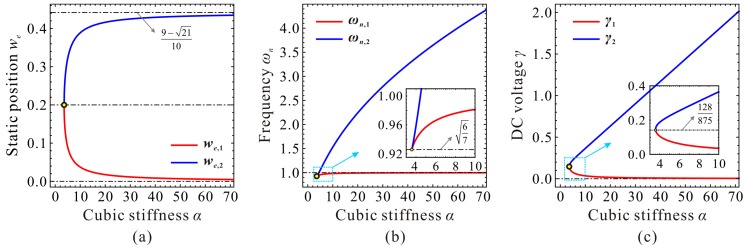
Variations of the nondimensional static position (**a**), fundamental frequency (**b**) and DC voltage (**c**) corresponding to point C_1_ and C_2_ versus cubic stiffness α.

**Figure 7 micromachines-09-00614-f007:**
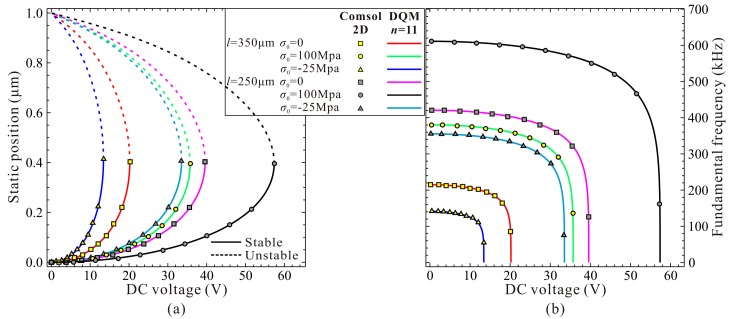
Static position (**a**) and fundamental frequency (**b**) for different beam length and residue stress.

**Figure 8 micromachines-09-00614-f008:**
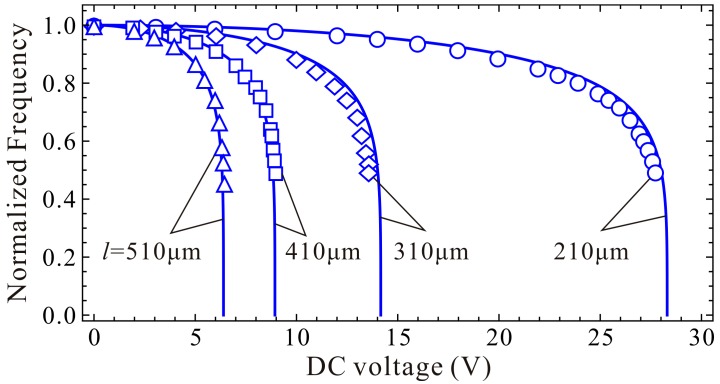
Comparison of the normalized fundamental frequencies between experiment result [39] (discrete point) and presented calculation with high-order differential quadrature method (DQM) using 11 grid points (solid line).

**Figure 9 micromachines-09-00614-f009:**
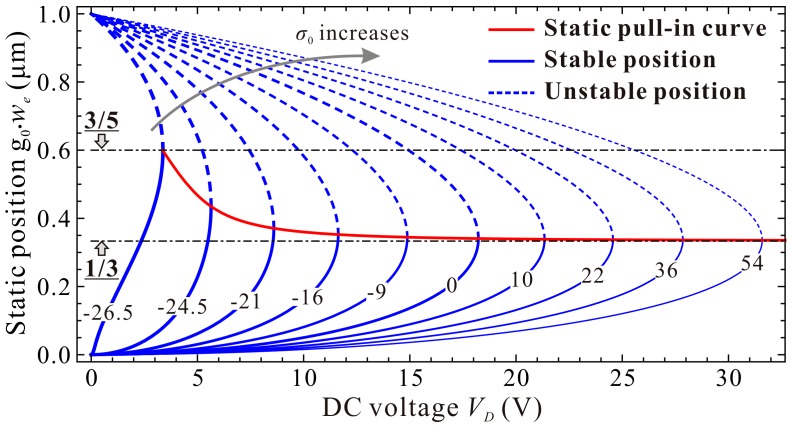
Qualitative property of the static position versus DC voltage for different residue stress (Unit: MPa).

**Figure 10 micromachines-09-00614-f010:**
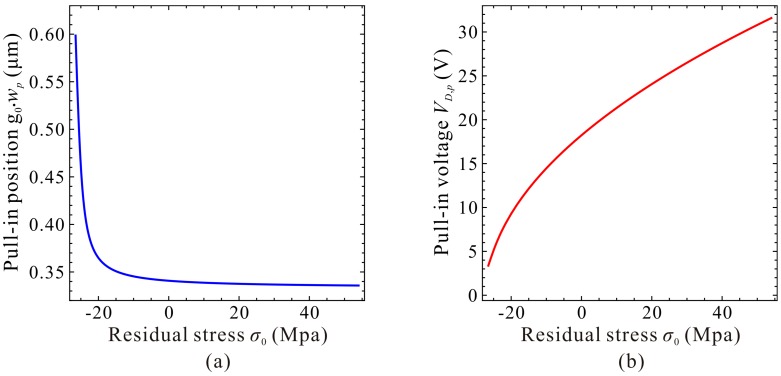
Qualitative property of the static pull-in position (**a**) and pull-in voltage (**b**) versus residual stress.

**Figure 11 micromachines-09-00614-f011:**
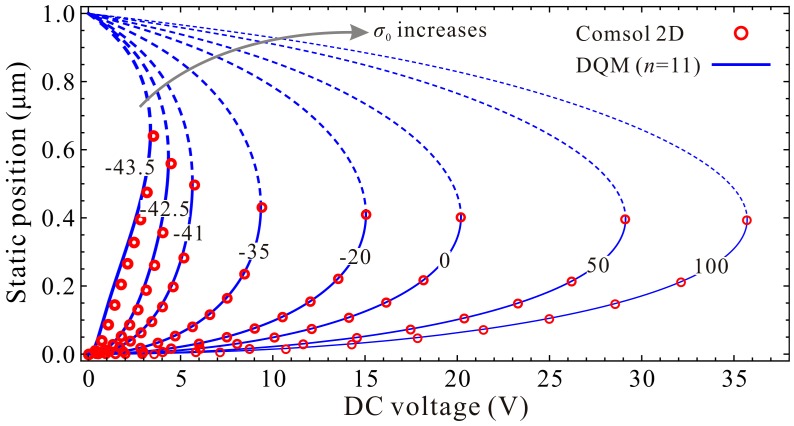
Quantitative property of the static position versus DC voltage for different residue stress (Unit: MPa).

**Figure 12 micromachines-09-00614-f012:**
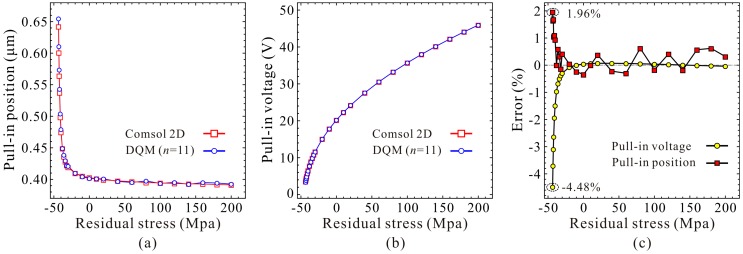
Quantitative property of the static pull-in voltage and pull-in position versus residual stress. Static pull-in position (**a**); Static pull-in voltage (**b**); Relative error (**c**).

**Figure 13 micromachines-09-00614-f013:**
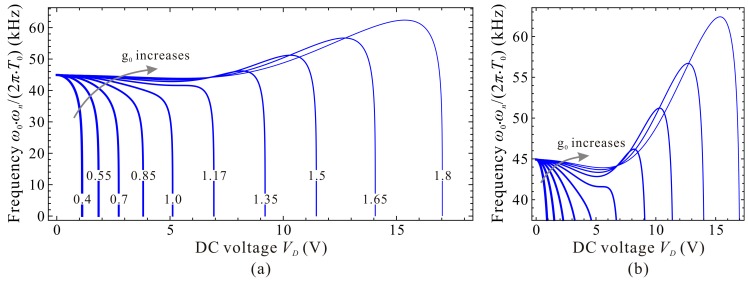
Qualitative property of the fundamental frequency versus DC voltage for different initial gap width (Unit: μm). global view (**a**); local view (**b**).

**Figure 14 micromachines-09-00614-f014:**
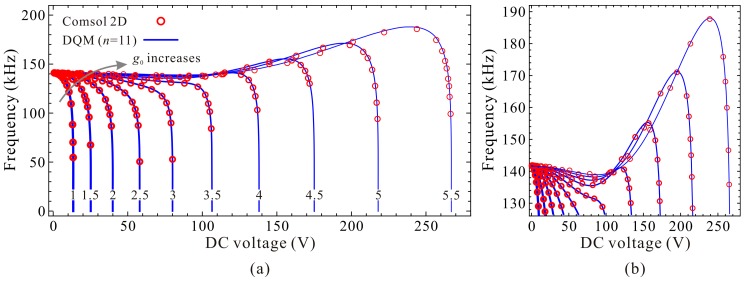
Quantitative property of the fundamental frequency versus DC voltage for different initial gap width (Unit: μm). Global view (**a**); Local view (**b**).

**Figure 15 micromachines-09-00614-f015:**
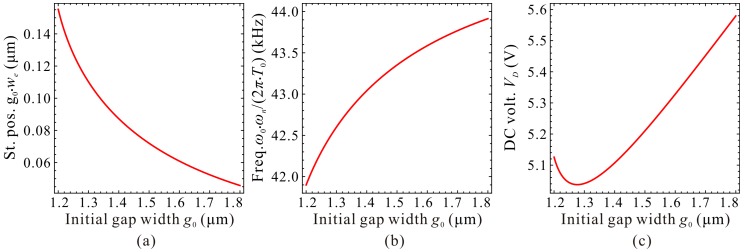
Qualitative property of the static position (St. pos.) (**a**), fundamental frequency (Freq.) (**b**) and DC voltage (DC volt.) (**c**) corresponding to point C_1_ on frequency curve versus initial gap width.

**Figure 16 micromachines-09-00614-f016:**
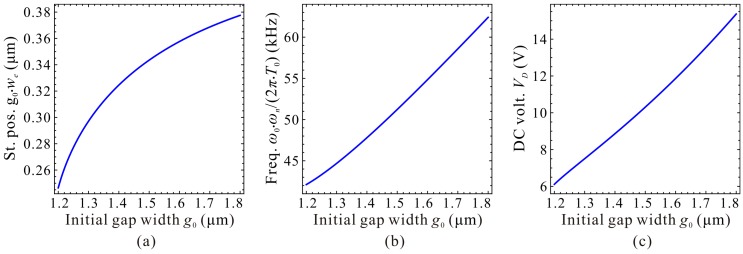
Qualitative property of the static position (St. pos.) (**a**), fundamental frequency (Freq.) (**b**) and DC voltage (DC volt.) (**c**) corresponding to point C_2_ on frequency curve versus initial gap width.

**Figure 17 micromachines-09-00614-f017:**
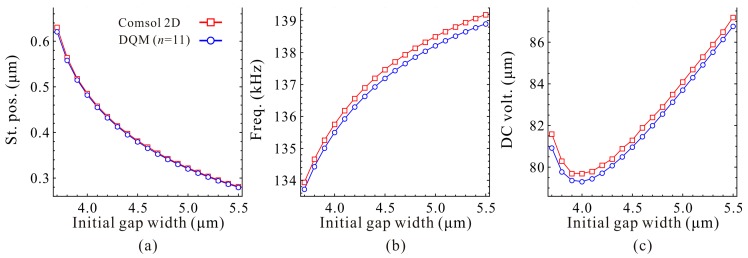
Quantitative property of the static position (St. pos.) (**a**), fundamental frequency (Freq.) (**b**) and DC voltage (DC volt.) (**c**) corresponding to point C_1_ on frequency curve versus initial gap width.

**Figure 18 micromachines-09-00614-f018:**
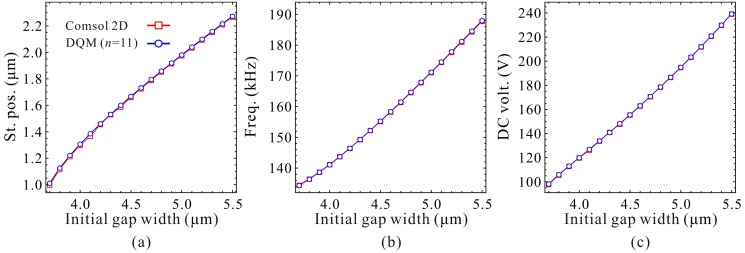
Quantitative property of the static position (St. pos.) (**a**), fundamental frequency (Freq.) (**b**) and DC voltage (DC volt.) (**c**) corresponding to point C_2_ on frequency curve versus initial gap width.

**Table 1 micromachines-09-00614-t001:** Theoretical thresholds of the static pull-in position and pull-in voltage.

Cubic Stiffness	Threshold of the Pull-in Position	Threshold of the Pull-in Voltage
α→∞	$w_{p}\to \frac{3}{5}$	$\gamma_{p}\to \infty$
α→0	$w_{p}\to \frac{1}{3}$	$\gamma_{p}\to \frac{4}{27}$

**Table 2 micromachines-09-00614-t002:** Static position, fundamental frequency and DC voltage corresponding to extreme point C_1_ and C_2_.

α	$w_{e,i}$	$\omega_{n,i}$	$\gamma_{i}$
$\alpha \;>\;\alpha_{c}$	$0<w_{e,1}<\frac{1}{5}$, $\frac{1}{5}<w_{e,2}<\frac{9-\sqrt{21}}{10}$	$\sqrt{\frac{6}{7}}<\omega_{n,1}<1$, $\sqrt{\frac{6}{7}}<\omega_{n,2}<+\infty$	$0<\gamma_{1}<\frac{128}{875}$, $\frac{128}{875}<\gamma_{2}<+\infty$
$\alpha =\alpha_{c}$	$w_{e,1}=w_{e,2}=\frac{1}{5}$	$\omega_{n,1}=\omega_{n,2}=\sqrt{\frac{6}{7}}$	$\gamma_{1}=\gamma_{2}=\frac{128}{875}$
$\alpha <\alpha_{c}$	—	—	—

**Table 3 micromachines-09-00614-t003:** Geometric and material parameters of a specific micro-electro-mechanical systems (MEMS) resonator [35].

Parameters	Values
Young’s modulus *E* (GPa)	169
Density *ρ_s_* (kg/m^3^)	2331
Poisson’s ratio *υ*	0.06
Width *b* (μm)	50
Thickness *h* (μm)	3
The absolute dielectric constant *ε*_0_ (F/m)	8.854187 × 10^−12^
The relative dielectric constant *ε_r_*	1
Initial gap width *g*_0_ (μm)	1

**Table 4 micromachines-09-00614-t004:** Static pull-in voltage of microresonator with different beam lengths and residual stress (Unit: V).

*l* (μm)	*σ*_0_ (MPa)	MEMCAD3D [38]	COMSOL 2D (Error: Λ %)	Results using DQM with different grid points *n* (Error: Λ %)				
				*n* = 5	*n* = 7	*n* = 9	*n* = 11	*n* = 13
*l* = 350	0	20.3	20.2 (−0.5)	18.2 (−10.3)	20.3 (0.0)	20.2 (−0.5)	20.2 (−0.5)	20.2 (−0.5)
	100	35.8	35.7 (−0.3)	39.6 (10.6)	36.4 (1.7)	35.7 (−0.3)	35.7 (−0.3)	35.7 (−0.3)
	−25	13.7	13.5 (−1.5)	5.1 (−62.8)	13.8 (0.7)	13.4 (−2.2)	13.4 (−2.2)	13.4 (−2.2)
*l* = 250	0	40.1	40.0 (−2.5)	35.7 (11.0)	39.8 (−0.7)	39.5 (−1.5)	39.6 (−1.2)	39.6 (−1.2)
	100	57.6	57.4 (−0.5)	60.8 (5.6)	58.0 (0.7)	57.3 (−0.5)	57.4 (−0.5)	57.4 (−0.5)
	−25	33.6	33.5 (−0.3)	25.9 (−22.9)	33.9 (0.9)	33.4 (−0.6)	33.5 (−0.3)	33.5 (−0.3)

with: $\Lambda =\frac{\mathrm{Calculated}\;\mathrm{results}-\mathrm{MEMCAD}\;\mathrm{results}}{\mathrm{MEMCAD}\;\mathrm{results}}\times 100\%$.

**Table 5 micromachines-09-00614-t005:** A comparision between the experimental and calculated resutls.

Beam Length (μm)	Fundamental Frequency (kHz)			Static Pull-in Voltage (V)		
	Experimental [39]	Calculated [9]	Present	Experimental [39]	Calculated [9]	Present
210	322.05	324.70	324.15	27.95	28.10	28.31
310	163.22	163.46	163.02	13.78	14.00	14.16
410	102.17	103.70	103.32	9.13	8.90	8.94
510	73.79	73.46	74.21	6.57	6.40	6.40

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
