# Peer review of "Qualitative Identification of the Static Pull-In and Fundamental Frequency of One-Electrode MEMS Resonators"

_micromachines, 2018, doi:10.3390/mi9120614_

Reviewer 1 Report

The paper is well-written and organized and could be interesting for the MEMS comunity. In any case, authors should address at least next major questions in order to clarify and identify the specific paper contribution and benefits of the analysis proposed:

·         Identify the advantage on using the DQM method for obtaining the 1DOF equation. It seems that the overall section 3 could be also performed by means of 1DOF from Garlekin method.

·         Authors states that the pull-in position and voltage are determined only by cubic stiffness (line 212), but in literature we find that the cubic stiffness and the linear one are geometrically related as K3/K1=12.2/16*width^2. In this sense, the linear stiffness also determines the pull-in.

·         Please explain why pull-in position increases as residual stress decreases (Fig. 9 and 10). Why the big difference between the bucling for the 1DOF obtained from DQM and the corresponding one for the Euler formulation?. Does it means that DQM is not as good as stated in the introduction?.

Reviewer 2 Report

Qualitative identification of the static pull-in and fundamental frequency of one-electrode MEMS resonators

Summary

This paper compare three different mathematical methods to study pull-in and fundamental frequency of one-electrode MEMS resonators. 1 degree of freedom DQM, higher order DQM, and 2D Comsol (FEM).

The paper reads well, presents the content in a way that is useful for the MEMS community as a whole. Specially, the experimental MEMS community in need of mathematical tools to solve experimental problems but that sometimes gets lost and afraid on too complex mathematics. This is a special virtue of this paper.

Major issues:

Structure the paper in a way that it is clearer when you talk about the “improved” 1-DOF DQM to make a qualitative analysis and when you talk about the general continuum equation higher order DQM to make a quantitative analysis. It is a bit confusing since both use the same discretization technique (DQM). Basically, structure the paper and specially the “System description” section to differentiate better between both DQMs. The one that you use for qualitative analysis and the one that you use (together with Comsol) for numerical analysis.

Minor issues:

Abstract: “However, nondimensional diagrams may exhibit some differences compared with dimensional ones, inducing some limitations of analytical conclusions. Perhaps, qualitative analyses combined with quantitative simulations are more suitable for the design and optimization of these microdevices.” This does not say anything important about the work presented in this paper. These are considerations that can go in the conclusions or discussion, but not in the abstract.

Page 4, Line 109: “the formulas to determine the static pull-in position, pull-in voltage and are derived.” Substitute the comma by the and.

Page 5, Line 140: “the formulas to determine the static pull-in position, pull-in voltage and are derived.” Substitute the comma by the and.

Page 6, Line 175: “However, from dynamic analysis perspective, Eq. (8) is not the simplest form” Why?

Page 6, Line 175: “Here, we improve Eq. (8) by reintroducing a time scale transformation xxx. Then, an improved nondimensional equation of motion can be obtained as” Why improved? Improved means better than? But better for which reasons? For which purpose? It can be better by being less time consuming in terms of computations but worse in terms of accuracy. Try to avoid subjective adjectives or at least specify why the use of the adjective.

Page 6, Line 176: Why the blue color?

Page 6, Line 189: What does it mean the equal symbol with a dot on top? Definition? Maybe just mention the meaning for the reader that is not aware of this.

Page 8, Line 229: “For simplicity and clarity, the finally solutions are listed in” Change finally by final.

Page 19, Line 454: “It is so exciting that all the dimensional diagrams with 1-DOF model show similar trends as those of quantitative results via DQM and COMSOL 2D simulations.” Rephrase the sentence. Exciting is not an appropriate word.

Page 19, Line 462: ““It becomes one research topic in our next work.” Unnecessary sentence. Please, remove.

Reviewer 3 Report

Qualitative identification of the static pull-in and fundamental frequency of one-electrode MEMS resonators

Jianxin Han, Lei Li, Gang Jin, Wenkui Ma, Jingjing Feng, Haili Jia, Dongmei Chang

The authors focus the study in one electrode MEMS resonators, arguing that from a quantitative analysis perspective, continuous models via Euler-Bernoulli beam theory or others seem to be more suitable for system analysis than time-varying capacitor model. I really appreciate the authors’ approach to the dynamic problem.

The manuscript is clear and very well written, and it could be of some interest for the community. I would like to see the paper published after considering some minor comments

1.     The authors argue the economic cost as shortcoming for the experimental realization of the study about the pull-in. However, the continuous advances in microelectronic industry make very cheap and easy to work with MEMS. You should reconsider this sentence.

2.     Do you think that this paper could be useful for practical realizations like for example dynamical switching? Could you please suggest some other possible applications

3.     How does the mode shape change as a function of the DC voltage? Does it have any influence on the resonance frequency? Please, say a word on it.

4.     How about higher-order modes? Although I know that this is not the scope of the paper I wonder about the resonance frequency of the second flexural mode, where there is a node (null displacement) at the middle point of the beam and there are two maxima with opposite displacement directions. Do you expect a frequency shift?

5.     Lines 393 and 394, the authors give the pull-in position with picometer accuracy. Experimentally talking there is no reason to that, especially while working with hundreds of microns length beams.

6.     Figs. 17 and 18. From my point of view, the error graph is not representative, I would suggest to represent the difference in between DQM and Comsol instead both graphs.

Author Response

Round  2

Reviewer 1 Report

Authors have addressed most of the reviewers comments and improved the paper. I think the  results could be interesting for MEMS community and the paper is suitable to be published in the present form.

Reviewer 2 Report

Line 107: Space between "resonance)of"

Line 117-118: This sentence is not clear "Different from the model by Najar et al. [22, 33], a dynamic equation of motion with 118 frequency normalization, is deduced by reusing a time scale transformation"

Reviewer 3 Report

The authors have addressed all my previous comments and concerns. Therefore, I would recommend the publication in the present form